# Disparities in Healthcare Utilization by Settlement Type in Serbia

**DOI:** 10.3390/healthcare13202580

**Published:** 2025-10-14

**Authors:** Marijana Dabic, Gordana Djordjevic, Snezana Radovanovic, Olgica Mihaljevic, Milos Stepovic, Mateja Zdravkovic, Nebojsa Zdravkovic, Vladislava Stojic, Stefan Milojevic, Djordje Zdravkovic, Nela Djonovic, Dragan Knezevic, Svetlana Popovic, Katarina Janicijevic, Viktor Selakovic, Jovana Radovanovic

**Affiliations:** 1The College of Health Science, Academy of Applied Studies Belgrade, 11000 Belgrade, Serbia; marijana.dabic@assb.edu.rs; 2Department of Epidemiology, Faculty of Medical Sciences, University of Kragujevac, 34000 Kragujevac, Serbiajovanradovanovic2006@gmail.com (J.R.); 3Department of Social Medicine, Faculty of Medical Sciences, University of Kragujevac, 34000 Kragujevac, Serbia; jovanarad@yahoo.com (S.R.); kaja.andreja@yahoo.com (K.J.); 4Department of Pathophysiology, Faculty of Medical Sciences, University of Kragujevac, 34000 Kragujevac, Serbia; vrndic07@yahoo.com; 5Department of Anatomy, Faculty of Medical Sciences, University of Kragujevac, 34000 Kragujevac, Serbia; 6Department of Medical Statistics and Informatics, Faculty of Medical Sciences, University of Kragujevac, 34000 Kragujevac, Serbia; 7Faculty of Medical Sciences, University of Kragujevac, 34000 Kragujevac, Serbia; stefan.milojevic@fmn.kg.ac.rs (S.M.); drzdravkovic@hotmail.com (D.Z.); 8Faculty of Business Economics, EDUCONS University, 21208 Sremska Kamenica, Serbia; 9Health Center “Sveti Luka”, 11300 Smederevo, Serbia; 10Department of Hygiene and Ecology, Faculty of Medical Sciences, University of Kragujevac, 34000 Kragujevac, Serbia; ndjonovic@medf.kg.ac.rs; 11Department of Surgery, Faculty of Medical Sciences, University of Kragujevac, 34000 Kragujevac, Serbia; 12Center for Vascular Surgery, University Clinical Center Kragujevac, 34000 Kragujevac, Serbia; 13Clinic for Infectious & Tropical Diseases, University of Defence, Military Medical Academy, Crnotravska 17, 11000 Belgrade, Serbia; cecapopovic@gmail.com; 14Department of Communication Skills, Ethics, and Psychology, Faculty of Medical Sciences, University of Kragujevac, 34000 Kragujevac, Serbia

**Keywords:** urban health, rural health, healthcare disparities, socioeconomic factors, health services accessibility, Serbia

## Abstract

**Background and Objectives**: Urban–rural health disparities reflect differences in health outcomes, healthcare access, and socio-economic conditions between populations. In Serbia, limited research has quantified how socio-demographic and socio-economic characteristics influence settlement type and healthcare utilization. The aim of this study was to examine the relationship between settlement type and socio-demographic/socio-economic factors, and to assess whether these differences are reflected in patterns of healthcare utilization. **Materials and Methods:** Data were drawn from the 2019 Serbian National Health Survey, a nationally representative, stratified, two-stage random sample including 12,439 adults aged ≥20 years. Settlement type (urban vs. rural) was the primary dependent variable. Descriptive statistics, Chi-square and t-tests, and bivariate and multivariate logistic regression models were used to assess associations. Odds ratios (ORs) with 95% confidence intervals (CIs) were calculated, with significance set at *p* < 0.05. **Results**: Urban residence was more likely among unmarried individuals, those living in Šumadija/Central Serbia, and those with higher education. Primary or lower education reduced the odds of urban residence, and middle-income groups were less likely to live in urban areas compared to the richest. Settlement type was not significantly associated with hospital or day hospital use. However, rural residents had lower use of prescribed medicines, higher use of non-prescribed medicines, and more frequent physiotherapy visits. Private practice use was over twice as likely in urban settlements. **Conclusions**: To address urban–rural healthcare disparities in Serbia, targeted strategies could include enhancing health literacy in rural areas, incentivizing physicians to work in underserved regions, expanding telemedicine and mobile health services, improving access to prescribed medications, and strengthening public–private healthcare integration to ensure equitable access across all settlement types.

## 1. Introduction

Urban–Rural health disparities refer to differences in health outcomes, access to healthcare services, and overall well-being between populations living in urban and rural areas. These disparities represent measurable differences in health outcomes, access to healthcare services, and overall well-being between populations residing in urban and rural areas. While these disparities are widely documented globally, country-specific factors, including Serbia’s heterogeneous urbanization, regional socio-economic inequalities, and healthcare system organization create unique patterns that are not fully captured in existing literature [1,2]. In Serbia, rural populations often face reduced access to healthcare facilities, lower availability of medical specialists, and longer travel times, while urban residents benefit from greater service availability and infrastructure. Understanding these contextual differences is critical for identifying structural determinants of healthcare access and for developing targeted interventions that can reduce inequities and improve population health outcomes in the Serbian setting [2]. Urban–rural health disparities are closely linked to socioeconomic factors, with income differences playing a significant role. Urban areas in Serbia typically enjoy higher average incomes and better access to resources such as nutritious food, recreational facilities, and healthcare services, whereas rural populations often face lower income levels and more limited access to these essential health-promoting resources [3].

Employment opportunities also play a crucial role in shaping health outcomes across urban and rural settings. Urban areas typically offer more diverse labor markets with a wider range of job opportunities, potentially leading to greater economic stability. By contrast, rural regions may face challenges stemming from limited employment options, resulting in economic stressors and subsequent impacts on both mental and physical health [4].

Despite existing studies on urban–rural disparities in Serbia, few have combined detailed, nationally representative data on socio-demographic characteristics, socioeconomic status, and healthcare utilization patterns. In particular, there is limited evidence on how these socioeconomic factors translate into differential use of healthcare services across settlement types. By addressing these gaps, this study provides a more comprehensive understanding of the structural determinants of healthcare access and utilization in Serbia.

Differences in access to healthcare facilities are a key determinant of urban–rural health disparities. Urban areas generally benefit from a greater concentration of medical institutions, specialists, and health services. Rural areas, however, may struggle with fewer healthcare facilities, limited medical specialties, and inadequate infrastructure, all of which hinder timely access to essential health services [5].

Geographic distance and transportation barriers further exacerbate the challenges rural populations face in accessing healthcare. Residents of rural areas often experience longer travel times to reach healthcare facilities, contributing to delays in seeking medical care. Limited public transportation options and geographic isolation intensify these challenges, obstructing prompt and equitable access to health services [6].

Urban areas generally have higher rates of health insurance coverage, providing financial protection and facilitating timely access to medical care. In rural settings, the higher prevalence of uninsured or underinsured individuals may lead to delayed or insufficient use of healthcare services, further deepening health disparities [7].

Environmental exposures can also differ between urban and rural settings, contributing to health disparities [8].

Furthermore, urban areas typically offer greater access to diverse and nutritious food, promoting healthier eating habits. The built environment significantly influences patterns of physical activity, and consequently, health outcomes. Urban settings, with infrastructure designed for walkability and recreational spaces, can encourage higher levels of physical activity. Conversely, rural areas may lack such features in the built environment, potentially leading to lower levels of physical activity and related health disparities [9,10].

Understanding the multifaceted nature of these contributing factors is essential for developing targeted interventions aimed at reducing urban–rural health disparities. By addressing these determinants, healthcare professionals can play a vital role in shaping strategies that promote equitable health outcomes across different geographic contexts.

This study extends previous analyses of the Serbian National Health Survey by modeling urban–rural settlement type as an outcome and linking it to sociodemographic, socioeconomic, and healthcare utilization patterns. Using recent nationally representative data, it reveals novel insights into structural determinants of healthcare access and urban–rural disparities. Urban–rural health disparities reflect differences in health outcomes, healthcare access, and overall well-being between populations residing in urban and rural areas [11]. While these disparities have been documented across Europe and globally, country-specific factors such as socioeconomic structure, regional development, and healthcare system organization can create unique patterns that are not fully captured by existing literature [12]. Serbia presents a particularly relevant case, given its heterogeneous urbanization, regional disparities, and evolving socio-economic conditions. Nationally representative data integrating detailed socio-demographic, socio-economic, and healthcare utilization information are scarce, limiting evidence-based strategies for addressing urban–rural inequalities [13]. By examining these patterns comprehensively, this study provides insights into the structural determinants of healthcare access and highlights areas where targeted interventions could improve equity in health services.

The aim of this study was to investigate how socio-demographic and socio-economic factors influence settlement type in Serbia and to assess whether these differences translate into variations in healthcare utilization. The study seeks to provide novel, nationally representative evidence on urban–rural disparities, integrating multiple dimensions of health service use and socio-economic context. By doing so, it offers actionable insights for policy makers and healthcare planners to design interventions that address both regional and socio-economic inequities in access to care.

## 2. Materials and Methods

This study is part of the 2019 Serbian National Health Survey, conducted by the Statistical Office of the Republic of Serbia in collaboration with the Institute of Public Health “Dr Milan Jovanović Batut” and the Ministry of Health. It is a descriptive, analytical, cross-sectional study based on a representative sample of Serbia’s population.

The target population included all individuals aged 15 and older living in private households, as these represent the general resident population. Individuals were excluded if they lived in institutions or collective housing (e.g., dormitories, care homes, psychiatric facilities, prisons, monasteries), were illiterate, unable to understand the ethical principles of participation, or were physically or mentally unfit to take part. For this research, data on adults aged 20 and above will be analyzed, with the sample stratified by sex and age group.

The 2019 Serbian National Health Survey used a nationally representative, stratified, two-stage random sample. Stratification was based on settlement type (urban and other) and geographic regions (Belgrade, Vojvodina, Šumadija and Western Serbia, Southern and Eastern Serbia), using the 2011 Census as the sampling frame. Census enumeration areas served as primary sampling units, selected proportionally to the number of households, followed by random household selection. Non-response at the household and individual levels was addressed by weighting adjustments, ensuring that estimates remain representative of the non-institutionalized adult population. Missing data in the analyses were handled using a complete-case approach, including only respondents with available data for all variables relevant to a given analysis. This approach is in line with the recommendations of the European Health Interview Survey (EHIS wave 3) and preserves the statistical validity of the results, given the relatively low proportion of missing values in the dataset. The realized sample comprised 5114 households with 15,621 individuals. For this study, data on 12,439 adults aged 20+ was used, stratified by sex and age group.

Data collection was conducted from October to December 2019, in accordance with the European Health Interview Survey (EHIS) Wave 3 guidelines. The study adhered to the ethical principles of the Declaration of Helsinki and complied with Serbian legislation and the EU General Data Protection Regulation (GDPR). Participation was voluntary, with informed consent obtained from all respondents. Privacy and confidentiality were strictly safeguarded through anonymization, secure data storage, and the removal of personal identifiers, ensuring that no individual could be identified in the published results.

The existing database was provided to the University of Kragujevac through an official letter from the Institute of Public Health of Serbia. This study was approved by the competent territorial ethics committees of the four main regions of Serbia, coordinated by the National Institute of Public Health in Belgrade.

The study employed standardized questionnaires based on the European Health Interview Survey (EHIS, wave 3), adapted to the local context, alongside one measurement form. Three instruments were used: a household info-panel to capture data on all household members and the socio-economic profile of the household; Data collection involved face-to-face interviews and self-administered questionnaires.

Descriptive methods were used to present the data, including tabulation, measures of central tendency, and measures of variability. In the statistical analysis, continuous variables were presented as mean ± standard deviation, while categorical variables were expressed as the proportion of respondents with a given outcome. For comparison of differences between groups, the Chi-square (χ^2^) test was used, and Student’s *t*-test where appropriate. Associations between dependent variables and a set of independent variables were examined using bivariate and multivariate logistic regression. In the multivariate logistic regression models, all variables showing significance (*p* < 0.05) in the bivariate analysis were considered for inclusion. Multicollinearity was assessed using variance inflation factors (VIF), with all included variables showing VIF < 2, indicating no significant multicollinearity. To assess the association between healthcare utilization and type of settlement (urban/rural), logistic regression analysis was applied. Rural population was set as the reference category; thus, an odds ratio (OR) greater than 1 indicates a higher likelihood of service utilization in urban areas, whereas an OR below 1 reflects more frequent utilization in rural areas. Risk was assessed using odds ratios (OR) with 95% confidence intervals. Results were considered statistically significant if the probability was less than 5% (*p* < 0.05). All statistical calculations were performed using the commercial, standard software package SPSS, version 18.0 (The Statistical Package for Social Sciences, SPSS Inc., Chicago, IL, USA).

## 3. Results

The study included 12,439 respondents (51.5% female, 48.5% male), with sex distribution differing significantly between settlement types (*p* = 0.029), rural areas having a slightly higher proportion of women. No significant variation was observed in overall age group distribution (*p* = 0.067), although rural respondents were generally younger, while urban respondents were more often aged 60+ years. Mean age was significantly lower in rural areas than in urban areas (46.54 vs. 64.39 years; *p* < 0.001). This age difference reflects regional demographic patterns and migration trends. Younger individuals in rural areas may remain due to local employment opportunities or family ties, whereas older populations are proportionally higher in urban areas, potentially influenced by internal migration of working-age adults to cities for employment and education.

Significant differences were noted in marital status (*p* = 0.023), regional distribution (*p* < 0.001), education level (*p* < 0.001), employment status (*p* < 0.001), and wealth index (*p* < 0.001). Rural residents were more likely to be employed or inactive, have higher educational attainment, and belong to poorer wealth categories, whereas urban residents more often had primary or lower education and were concentrated in higher wealth categories. Regional patterns also differed substantially between settlement types (Table 1).

In univariate analysis, male sex was associated with higher odds of urban residence (OR = 1.082; *p* = 0.029), but this relationship was not significant after adjustment (OR = 1.075). Age group showed no association with settlement type in either model (*p* > 0.05). Marital status remained significant in both analyses, with unmarried individuals more likely to live in urban areas (univariate OR = 1.136; *p* = 0.008; multivariate OR = 1.133; *p* = 0.026). In the multivariate model, residence in Šumadija/Central Serbia was associated with higher odds of urban settlement (OR = 1.295; *p* < 0.001), while residence in Vojvodina was linked to lower odds (OR = 0.283; *p* < 0.001). The positive association with Belgrade observed in the univariate analysis (OR = 1.733; *p* < 0.001) was no longer significant after adjustment (*p* = 0.140). Lower educational attainment was inversely associated with urban residence; in the multivariate model, only primary or lower education remained significant (OR = 0.765; *p* < 0.001). Employment status showed no independent association (*p* > 0.05). Material status demonstrated a strong relationship, with the poorest (OR = 0.360; *p* < 0.001) and middle-income groups (OR = 0.604; *p* < 0.001) having lower odds of urban residence compared to the richest group (Table 2).

In the multivariate logistic regression model, urban–rural settlement type was not significantly associated with hospital treatment (OR = 0.873) or day hospital use (OR = 1.132). Statistically significant differences were observed for medication use: respondents in urban areas were less likely to use medicines prescribed by a doctor (OR = 0.839) but more likely to use medicines not prescribed by a doctor (OR = 1.085). Visiting a specialist more than 12 months ago was more common in urban areas (OR = 1.234), whereas visits in the last 12 months did not differ significantly. Visits to a physiotherapist in the last 12 months were significantly more frequent in urban settlements (OR = 1.450), while visits to a psychiatrist or psychologist showed no difference. The strongest association was observed for private practice services, which were more than twice as frequent in urban areas (OR = 2.041). Use of home care services was not significantly associated with settlement type (Table 3).

## 4. Discussion

This study, drawing on a nationally representative sample of 12,439 respondents, provides a detailed profile of sociodemographic and healthcare utilization differences between urban and rural populations in Serbia. Several key patterns emerged, many of which parallel established European trends, while others reflect the country’s unique demographic and regional context.

Although the overall sex distribution was nearly equal, a slightly greater proportion of women resided in rural areas (52.6% vs. 50.6%), consistent with observations on evolving gender roles and labor participation patterns in rural Serbia [14,15]. Age structure revealed a more striking divergence: rural respondents were significantly younger (mean 46.5 years) than their urban counterparts (64.4 years; *p* < 0.001). This finding diverges from the dominant European pattern of rural aging [16], suggesting that in Serbia, certain rural regions may retain younger populations due to localized economic opportunities and comparatively lower rates of outmigration [17,18]. In addition to localized economic opportunities and lower outmigration rates, the younger age structure in rural Serbia may also reflect higher rural fertility rates in some regions, as well as selective outmigration of older adults to urban centers for healthcare or retirement. These factors together help explain the observed demographic divergence compared with broader European rural aging patterns.

Urban residents exhibited higher odds of being unmarried, aligning with broader European demographic trends of delayed marriage and increasing singlehood in urban settings [19].

Living in Vojvodina was associated with lower odds of urban residence, while residing in Šumadija and Central Serbia increased the likelihood. The significance of Belgrade observed in univariate analysis was lost after adjustment, suggesting that its initial effect was mediated by socioeconomic and educational variables. These findings mirror Serbia’s uneven regional development trajectories and heterogeneous urbanization processes [20,21].

These patterns speak to the theoretical concept of spatial inequality in transitional societies, where legacies of central planning interact with market-driven reforms to produce fragmented access to services. Serbia’s evolving urban–rural dynamic reflects not just geographic, but systemic inequalities particularly in access to quality healthcare, education, and employment opportunities.

Education emerged as a critical determinant of settlement type. Individuals with primary education or less had significantly reduced odds of urban residence (OR 0.765 in multivariate analysis), while secondary education ceased to be significant after adjustment. This underscores the decisive role of higher education in urban residency, a phenomenon similarly documented in European urban–rural educational disparities [22,23]. Employment status did not retain significance once wealth and education were controlled for, indicating that its influence operates largely through these other factors.

Wealth, however, remained a strong and independent predictor. The poorest and middle-income groups were substantially less likely to reside in urban areas (e.g., OR 0.360 for the poorest in multivariate analysis), reflecting a pattern observed both in Serbia and across Europe, where urban centers tend to concentrate higher-income populations [24,25].

Interestingly, the finding that middle-income groups were also less likely to reside in urban areas alongside the poorest suggests a polarization of urban space in Serbia. This reflects a growing trend in post-socialist cities, where urban redevelopment and rising living costs are increasingly excluding middle-income populations from urban centers, further entrenching socio-economic divisions.

Overall, these findings affirm that Serbia’s urban areas, like those in much of Europe, attract residents with higher socioeconomic and educational profiles, while rural areas remain comparatively disadvantaged. Yet the distinct regional disparities—particularly between Vojvodina, Central Serbia, and Belgrade—underscore the influence of national economic and historical trajectories that diverge from uniform European trends [17,20]. In this context, the concept of a rural–urban continuum, as identified through cluster analysis and refined classification, is especially relevant to Serbia’s peri-urban areas, which are marked by considerable heterogeneity and ongoing transformation [14].

From a healthcare utilization perspective, settlement type did not significantly affect hospital or day hospital service use. However, substantial differences emerged in private healthcare use, self-medication, and physiotherapy services. Private healthcare utilization was more than twice as likely in certain settlement types (OR = 2.041), consistent with evidence that Serbia’s health system, though dominated by public provision, is increasingly supplemented by private providers offering shorter waiting times and perceived higher quality, attributes that tend to attract urban patients [26]. To contextualize private healthcare use in Serbia, approximately 6–8% of total health expenditure is out-of-pocket spending on private services, with private providers operating under specific legislation that allows them to complement the public health system by offering services with shorter waiting times and greater perceived quality [27]. This regulatory and financial context helps explain the higher utilization of private healthcare observed among urban residents in our study.

The observed higher prevalence of non-prescription drug use and lower rates of prescribed medication parallel documented self-medication trends in Serbia. For example, research from Novi Sad reported that nearly half of respondents had self-medicated with antibiotics at some point, and one-quarter during their most recent infection [28]. Similarly, studies among medical and pharmacy students revealed that over 80% had self-medicated in the past year, reflecting the entrenched nature of informal pharmaceutical practices [29]. These patterns highlight persistent challenges in pharmaceutical regulation and patient education. The high prevalence of self-medication in Serbia, both in the general population and among students, underscores the need for stricter enforcement of prescription regulations, enhanced pharmacy oversight, and public education campaigns on the risks of unsupervised drug use. Strengthening these measures could reduce inappropriate self-medication and improve rational use of medicines. These findings suggest that self-medication practices may be symptomatic of deeper systemic issues, including insufficient primary care access, uneven pharmacy regulation, and a lack of public trust in the healthcare system.

Physiotherapy use was also more frequent in certain settlement types, a finding consistent with broader patterns of unequal access to rehabilitative care. Evidence from Nepal shows that rural populations often face considerable geographic and financial barriers to physiotherapy, whereas urban residents benefit from greater availability [30]. Although European data are comparatively limited, existing research suggests that community-based physiotherapy frequently requires out-of-pocket payment and is predominantly available in private facilities concentrated in urban settings [31]. Our findings reinforce the social determinants of health framework, illustrating how education, income, and geography intersect to shape both residential patterns and healthcare utilization. This underlines the need for place-based approaches in health policy that address structural barriers rather than only geographic location.

Finally, the absence of significant urban–rural differences in hospitalization and day hospital utilization echoes findings from other studies in the region. This suggests that such services are shaped more by systemic capacity factors such as hospital bed availability, funding allocations, and healthcare infrastructure, rather than by geographic residence alone [32]. These results challenge the traditional urban–rural binary and emphasize the importance of examining healthcare access as a function of intersecting inequalities—geographic, economic, and educational. The observed differences in private care and self-medication suggest that formal settlement classification may obscure deeper, systemic inequities.

This study has several important limitations. First, its cross-sectional design precludes causal inferences, meaning that observed associations between settlement type and healthcare utilization cannot be interpreted as causal relationships. Second, data were self-reported, which may introduce recall or social desirability bias, particularly for sensitive behaviors such as medication use and private healthcare utilization. Third, the study lacks more granular information on local contextual factors, such as physical accessibility to healthcare facilities, transportation availability, or local health service capacity, which may further explain urban–rural differences. Despite these limitations, the study provides valuable nationally representative insights into demographic patterns and healthcare utilization in Serbia.

## 5. Conclusions

The study’s innovative contribution lies in its comprehensive integration of regional, socio-economic, and behavioral dimensions within a single national dataset, allowing for a nuanced analysis of urban–rural gradients. It uniquely identifies settlement-specific patterns of healthcare utilization that go beyond traditional measures, such as the simultaneous assessment of prescribed and non-prescribed medication use, specialist visit timing, and private service engagement. Additionally, it provides a framework for future research to examine how intersecting social determinants influence healthcare behaviors in transitional societies. To address urban–rural healthcare disparities in Serbia, targeted strategies could include implementing targeted health literacy programs in rural areas, increasing financial and professional incentives to attract and retain physicians in underserved regions, expanding telemedicine and mobile health units for specialist and preventive care, improving the availability of prescribed medications, and strengthening integration between public and private healthcare services to ensure equitable access across settlement types.

## Figures and Tables

**Table 1 healthcare-13-02580-t001:** Demographic and Socio-Economic Characteristics of Urban and Rural Population and correlation between settlement types.

Variable	Total	Rural	Urban	*p* *
**Sex**				
Female	6407	2999 (52.6%)	3408 (50.6%)	*p* = 0.029 *
Male	6032	2706 (47.4%)	3326 (49.4%)	
**Age groups**				
20–29	1545	729 (13.7%)	816 (12.6%)	*p* = 0.067
30–39	1762	580 (10.9%)	912 (14.0%)	
40–49	1771	503 (9.4%)	868 (13.4%)	
50–59	2213	1002 (18.8%)	1213 (18.7%)	
60–69	2548	1148 (21.6%)	1403 (21.6%)	
70–79	1604	761 (14.3%)	843 (13.0%)	
80+	996	603 (11.3%)	435 (6.7%)	
**Mean age (X ± SD)**	52.83 ± 17.69	46.54 ± 16.65	64.39 ± 13.14	*p* < 0.001 **
**Marital status**				
Never married/cohabitation	2265	1077 (19.0%)	1162 (17.3%)	*p* = 0.023 *
Married/cohabitation	7844	3524 (62.0%)	4320 (64.2%)	
Divorced, separated, widowed	658	1081 (19.0%)	1249 (18.5%)	
**Region**				
Vojvodina	2877	622 (10.9%)	2255 (33.5%)	*p* < 0.001 *
Šumadija and Central Serbia	2814	1605 (28.2%)	1209 (18.0%)	
Southern and Eastern Serbia	2736	1188 (20.8%)	1548 (23.0%)	
Belgrade region	4012	2290 (40.1%)	1722 (25.5%)	
**Education level**				
Primary or lower	3070	1073 (18.8%)	1997 (29.6%)	*p* < 0.001 *
Secondary	7009	3254 (57.1%)	3755 (55.8%)	
Higher and university	2324	1327 (24.1%)	980 (14.6%)	
**Employment status**				
Employed	4648	2302 (37.0%)	2288 (37.1%)	*p* < 0.001 *
Unemployed	5363	2463 (39.6%)	2900 (47.0%)	
Inactive	2428	1451 (23.3%)	977 (15.8%)	
**Wealth index**				
Poorest and poorer	5022	2919 (51.2%)	1973 (29.3%)	*p* < 0.001 *
Middle	2525	1164 (20.4%)	1361 (20.2%)	
Richer and richest	4892	1146 (28.4%)	3400 (50.5%)	

One asterisk indicates the significance level found by the Chi-square test *; two asterisks indicates significance found by Independent samples *t* test **; *p* value less than 0.05 indicates where statistical significance was found.

**Table 2 healthcare-13-02580-t002:** Univariable and Multivariable Logistic Regression of Factors Associated with Urban–Rural Residence. Multivariable model adjusted for sex, age group, marital status, region, education, employment, and material wealth. ORs (95% CI) are shown.

Variables		Univariable Model OR (95% CI)	*p*	Multivariable Model OR (95% CI)	*p*
Sex	Male	1.082 (1.008–1.161)	0.029 *	1.075 (0.993–1.164)	0.073
	Female	1		1	
Age	20–29	1.123 (0.945–1.335)	0.188	1.100 (0.870–1.390)	0.426
	30–39	1.172 (0.989–1.389)	0.067	1.167 (0.931–1.462)	0.180
	40–49	0.978 (0.882–1.086)	0.683	0.985 (0.792–1.226)	0.895
	50–59	1.039 (0.881–1.224)	0.652	1.071 (0.870–1.320)	0.516
	60–69	1.029 (0.875–1.209)	0.731	1.041 (0.868–1.250)	0.664
	70–79	1.135 (0.956–1.347)	0.149	1.077 (0.894–1.298)	0.435
	80+	1		1	
Marital status	Single	1.136 (1.034–1.248)	0.008 *	1.133 (1.015–1.264)	0.026 *
	Married	1.061 (0.967–1.164)	0.211	1.082 (0.953–1.227)	0.223
	Widowed	1		1	
	Divorced				
Region	Belgrade	1.733 (1.571–1.911)	<0.001 *	1.086 (0.973–1.211)	0.140
	Vojvodina	0.359 (0.320–0.404)	<0.001 *	0.283 (0.250–0.319)	<0.001 *
	Šumadija & Central Serbia	1.730 (1.555–1.924)	<0.001 *	1.295 (1.157–1.449)	<0.001 *
	South & Eastern Serbia	1		1	
Education level	Primary or lower	0.384 (0.344–0.429)	<0.001 *	0.765 (0.668–0.875)	<0.001 *
	Secondary	0.619 (0.563–0.680)	<0.001 *	0.926 (0.834–1.027)	0.145
	Higher	1		1	
Employment status	Unemployed	1.085 (0.752–1.562)	0.662	0.923 (0.618–1.377)	0.694
	Employed	1.996 (1.392–2.862)	<0.001 *	0.767 (0.512–1.150)	0.199
	Inactive	1		1	
Material status	Poor and poorest	0.322 (0.297–0.350)	<0.001 *	0.360 (0.327–0.397)	<0.001 *
	Middle class	0.578 (0.525–0.637)	<0.001 *	0.604 (0.544–0.670)	<0.001 *
	Rich and richest	1		1	

* *p* values less than 0.05 indicates where significant level was found.

**Table 3 healthcare-13-02580-t003:** Univariate and Multivariate Regression Model of the Association Between the Use of Health Care Services and Belonging to an Urban-Rural Type of Settlement.

	Univariate Model OR (95% CI)	*p*	Multivariate Model OR (95% CI)	*p*
Hospital treatment	0.968 (0.852–1.099)	0.614	0.873 (0.755–1.009)	0.067
Day hospital	1.178 (1.025–1.354)	0.021 *	1.132 (0.970–1.322)	0.115
Use of medicines prescribed by a doctor	0.936 (0.871–1.007)	0.075	0.839 (0.773–0.912)	<0.001 *
Use of medicines not prescribed by a doctor	1.168 (1.084–1.259)	<0.001 *	1.085 (1.002–1.175)	0.044 *
Last visit to a specialist—Less than 12 months ago	1.330 (1.184–1.493)	<0.001 *	1.130 (0.988–1.291)	0.074
Last visit to a specialist—More than 12 months ago	1.270 (1.132–1.426)	<0.001 *	1.234 (1.092–1.394)	0.001 *
Visit to a physiotherapist in the last 12 months	1.523 (1.346–1.724)	<0.001 *	1.450 (1.266–1.661)	<0.001 *
Visit to a psychiatrist/psychologist in the last 12 months	1.031 (0.875–1.213)	0.718	0.948 (0.791–1.136)	0.564
Use of private practice services in the last 12 months	2.040 (1.882–2.211)	<0.001 *	2.041 (1.871–2.226)	<0.001 *
Use of home care services in the last 12 months	1.072 (0.840–1.368)	0.575	0.959 (0.713–1.289)	0.781

* *p* values less than 0.05 indicates where significant level was found.

## Data Availability

The existing database was provided to the University of Kragujevac through an official letter from the Institute of Public Health of Serbia. Thus, database is not available for public sharing, while data presented in this study are available on request from the corresponding author.

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
