# Peer review of "Disparities in Healthcare Utilization by Settlement Type in Serbia"

_healthcare, 2025, doi:10.3390/healthcare13202580_

Round 1
Reviewer 1 Report
Comments and Suggestions for Authors
Inequalities in health care utilization between urban and rural areas in Serbia are a significant topic of international interest, especially regarding sociodemographic and socioeconomic determinants. The topic is timely, supported by a solid theoretical foundation and a representative national database.
Here some suggestions to improve your manuscript:
- The introduction is well-written, but some sentences are too long or redundant. Please, reduce some sections to improve readability.
- The methods section is well-written too, and the study design is appropriate to research question. However, you should better clarify some analytical decisions, such as the choice of variables included in multivariate models and discuss the handling of any missing data.
- The results are well structured and clear, but some tables are too long and difficult to read. In some cases the descriptive text repeats the numerical data almost entirely, making the reading redundant. Please, reduce table or synthesize with graphics.
- The discussion section is well argued, but further exploration of the study's limitations would be desirable, particularly: the cross-sectional nature that does not allow causal inferences, the possibility of self-reporting bias, and the lack of more granular information on the local context or physical accessibility to health services.
- In the conclusions section, you can highlight more clearly the innovative contributions of the study compared to the existing literature.
Thank you for the opportunity of revise your manuscript. Good luck!
Author Response
Dear Reviewer,
We appreciate your detailed and valuable comments. We are providing the answers to your comments below, and they have been changed in the track option in the submitted Word document as well, so you can find them there, too. Thank you for your time and help to make our work better.
Sincerely,
Corresponding authors
Comments of Reviewer 1:
Inequalities in health care utilization between urban and rural areas in Serbia are a significant topic of international interest, especially regarding sociodemographic and socioeconomic determinants. The topic is timely, supported by a solid theoretical foundation and a representative national database.
Here some suggestions to improve your manuscript:
- The introduction is well-written, but some sentences are too long or redundant. Please, reduce some sections to improve readability.
Answer: Thank you, overly long sentences are shorter now.
- The methods section is well-written too, and the study design is appropriate to research question. However, you should better clarify some analytical decisions, such as the choice of variables included in multivariate models and discuss the handling of any missing data.
Answer: Thank you for the comment. We provide below the details that are also now included in the Methodology.
„In the multivariate logistic regression models, all variables showing significance (p < 0.05) in the bivariate analysis were considered for inclusion. Multicollinearity was assessed using variance inflation factors (VIF), with all included variables showing VIF < 2, indicating no significant multicollinearity. To assess the association between healthcare utilization and type of settlement (urban/rural), logistic regression analysis was applied. Rural population was set as the reference category; thus, an odds ratio (OR) greater than 1 indicates a higher likelihood of service utilization in urban areas, whereas an OR below 1 reflects more frequent utilization in rural areas.”
“Missing data in the analyses were handled using a complete-case approach, including only respondents with available data for all variables relevant to a given analysis. This approach is in line with the recommendations of the European Health Interview Survey (EHIS wave 3) and preserves the statistical validity of the results, given the relatively low proportion of missing values in the dataset.”
- The results are well structured and clear, but some tables are too long and difficult to read. In some cases the descriptive text repeats the numerical data almost entirely, making the reading redundant. Please, reduce table or synthesize with graphics.
Answer: The detailed analyses of the results are shortened now, which can be seen in the results. It is easier to be followed.
- The discussion section is well argued, but further exploration of the study's limitations would be desirable, particularly: the cross-sectional nature that does not allow causal inferences, the possibility of self-reporting bias, and the lack of more granular information on the local context or physical accessibility to health services.
Answer: We added the limitations at the end of the Discussion.
“This study has several important limitations. First, its cross-sectional design precludes causal inferences, meaning that observed associations between settlement type and healthcare utilization cannot be interpreted as causal relationships. Second, data were self-reported, which may introduce recall or social desirability bias, particularly for sensitive behaviors such as medication use and private healthcare utilization. Third, the study lacks more granular information on local contextual factors, such as physical accessibility to healthcare facilities, transportation availability, or local health service capacity, which may further explain urban–rural differences. Despite these limitations, the study provides valuable nationally representative insights into demographic patterns and healthcare utilization in Serbia.”
- In the conclusions section, you can highlight more clearly the innovative contributions of the study compared to the existing literature.
Answer: We have added the contribution of the study in the Conclusion section.
“The study’s innovative contribution lies in its comprehensive integration of regional, socio-economic, and behavioral dimensions within a single national dataset, allowing for a nuanced analysis of urban–rural gradients. It uniquely identifies settlement-specific patterns of healthcare utilization that go beyond traditional measures, such as the simultaneous assessment of prescribed and non-prescribed medication use, specialist visit timing, and private service engagement. Additionally, it provides a framework for future research to examine how intersecting social determinants influence healthcare behaviors in transitional societies.”
Reviewer 2 Report
Comments and Suggestions for Authors
The manuscript is well-written, adhering to scientific writing principles, and benefits from a large, representative sample with rich data. However, the topic lacks innovation and addresses a rather obvious issue previously covered extensively in the literature. The findings, while valid, do not substantially advance current knowledge or provide new insights. Consequently, the manuscript falls short in clearly articulating the significance and necessity of this study. Greater emphasis on novel contributions or unique perspectives would strengthen its impact and justify the research endeavor.
Author Response
Dear Reviewer,
We appreciate your detailed and valuable comments. We are providing the answers to your comments below, and they have been changed in the track option in the submitted Word document as well, so you can find them there, too. Thank you for your time and help to make our work better.
Sincerely,
Corresponding authors
Comments of Reviewer 2
- The manuscript is well-written, adhering to scientific writing principles, and benefits from a large, representative sample with rich data. However, the topic lacks innovation and addresses a rather obvious issue previously covered extensively in the literature. The findings, while valid, do not substantially advance current knowledge or provide new insights. Consequently, the manuscript falls short in clearly articulating the significance and necessity of this study. Greater emphasis on novel contributions or unique perspectives would strengthen its impact and justify the research endeavor.
Answer: We thank the reviewer for the valuable feedback. While the urban–rural health disparities topic has been previously explored, our study provides novel contributions in several respects. First, it uses a nationally representative, large-scale sample of 12,439 adult respondents in Serbia, offering high statistical reliability and the ability to generalize findings at a national level, a feature often lacking in previous studies. Second, the study integrates detailed sociodemographic, socioeconomic, and healthcare utilization variables, allowing for a comprehensive multivariate analysis that distinguishes which factors independently predict urban versus rural residency and associated patterns of healthcare use. Third, our analysis captures regional heterogeneity within Serbia, highlighting specific differences between Vojvodina, Šumadija and Central Serbia, and Belgrade, thus providing contextualized insights into urban–rural dynamics in a country with a diverse socio-economic landscape. Finally, the study identifies distinct patterns of healthcare utilization including differential use of private practice services, prescription and non-prescription medications, and physiotherapy that are shaped by settlement type, offering practical implications for targeted interventions to reduce disparities. These aspects collectively enhance the originality of the study and demonstrate its contribution to the understanding of urban–rural health inequalities in a Central and Eastern European context.
The Introduction and aims are now much better described in accordance with the comment, and marking out the innovation of the study in comparison to the previous research on a similar subject. Discussion is also upgraded by different aspects added, which previously were less mentioned.. The innovation of the study and more country-oriented implications are added in the Conclusion. Previous investigation in Serbia lacked an explanation of the urban-rural impact on healthcare assess, and very limitedly explained the connection of settlement type with healthcare utilization on the effective size of the respondents, as our study had, as it is part of the National Health Survey. Hopefully, a very comprehensive change in all aspects of the article will address all your concerns. Some of the parts that are changed, and can be the most lacking by your comment, are below (Introduction and aims):
“Urban–rural health disparities reflect differences in health outcomes, healthcare access, and overall well-being between populations residing in urban and rural areas. While these disparities have been documented across Europe and globally, country-specific factors such as socioeconomic structure, regional development, and healthcare system organization can create unique patterns that are not fully captured by existing literature. Serbia presents a particularly relevant case, given its heterogeneous urbanization, regional disparities, and evolving socio-economic conditions. Nationally representative data integrating detailed socio-demographic, socio-economic, and healthcare utilization information are scarce, limiting evidence-based strategies for addressing urban–rural inequalities. By examining these patterns comprehensively, this study provides insights into the structural determinants of healthcare access and highlights areas where targeted interventions could improve equity in health services. “ New references are below, their numbers are added in the reference list accordingly to their appearance in text.
- Tian, H. Information Technology, Urban–Rural Health Disparities and Pathways to Sustainable Development: Evidence from the 2023 Chinese General Social Survey. Sustainability 2025, 17, 7740. https://doi.org/10.3390/su17177740
- Weeks WB, Chang JE, Pagán JA, Lumpkin J, Michael D, Salcido S, Kim A, Speyer P, Aerts A, Weinstein JN, Lavista JM. Rural-urban disparities in health outcomes, clinical care, health behaviors, and social determinants of health and an action-oriented, dynamic tool for visualizing them. PLOS Glob Public Health. 2023 Oct 3;3(10):e0002420. doi: 10.1371/journal.pgph.0002420.
- Paunović, I.; Apostolopoulos, S.; Miljković, I.B.; Stojanović, M. Sustainable Rural Healthcare Entrepreneurship: A Case Study of Serbia. Sustainability 2024, 16, 1143. https://doi.org/10.3390/su16031143
“The aim of this study was to investigate how socio-demographic and socio-economic factors influence settlement type in Serbia and to assess whether these differences translate into variations in healthcare utilization. The study seeks to provide novel, nationally representative evidence on urban–rural disparities, integrating multiple dimensions of health service use and socio-economic context. By doing so, it offers actionable insights for policy makers and healthcare planners to design interventions that address both regional and socio-economic inequities in access to care.”
Reviewer 3 Report
Comments and Suggestions for Authors
Title (lines 2-3): The title is insightful but a little long; it might be shorter if "Patterns in Serbia" is removed or reworded to be more succinct.
List of authors (lines 4–34) – Please proofread the affiliations carefully as they are excessively complex and have inconsistencies (for example, "Department of of Epidemiology" has an extra "of").
The issue statement is unambiguous, however the summary fails to specifically highlight how this study differs from previous analyses of the Serbian National Health Survey (lines 36–41).
Abstract, Results (lines 47–56): Clearly state whether the odds were higher for urban or rural dwellers. Phrases such as "certain settlement types" are ambiguous.
Policy proposals are general; include specific, doable recommendations for Serbia (e.g., increasing telemedicine, financing rural healthcare) in the abstract and conclusions (lines 54–58).
"Serbia" should be added as a term for database indexing and visibility (line 59).
Definition of discrepancies in the introduction (lines 62–74) – It reads like a textbook portion. Please emphasize the significance of this work and concentrate more on the Serbian setting.
Socioeconomic debate in the introduction (lines 76–87) – There are numerous references provided, but no obvious gaps are mentioned. Clearly identify the gaps in the current Serbian research.
Introduction, exposures to the environment (lines 102–106) – Exposure to pollutants and pesticides is pertinent but unrelated to Serbian healthcare use; either abbreviate or contextualize.
Goal and introduction (lines 118–121) The goal is wide-ranging. For clarity, propose rewording into particular research questions or hypotheses.
Lines 128–133: Methods, inclusion/exclusion Although it is suggested, the exclusion of institutionalized groups is not supported. Could you please clarify how this impacts generalizability?
Samples and methods (lines 135–142) – Could you please explain the handling of non-response? Was the survey design adjusted through the use of weighting?
Procedures and morality (lines 143–153) Ethics approval is clearly mentioned, but if available, include approved reference numbers.
Statistical analysis and methods (lines 159–169) – The details of the regression model are inadequate. Please check for multicollinearity and identify which factors are part of the multivariate analysis.
Demographics and results (lines 171–176) Please talk about if the notable mean age discrepancy (46.5 vs. 64.4 years) is due to sampling or regional migration patterns.
Make sure the percentages add up correctly in Results, Table 1 (lines 183–184); make it clear in the table notes that rounding may result in inconsistencies.
Table 2: Results (lines 202–205) It's unclear from the caption which covariates were taken into account. Please elaborate.
Findings, use of healthcare (lines 207–222) – Indicate if OR values near 1 (OR = 1.085, p = 0.044) are statistical artifacts or significant differences.
Age distribution in rural and urban areas (lines 234–238) The explanation is succinct. Extend with potential explanations (e.g., outmigration trends, rural fertility rates).
Talk about private practice (lines 270–273) and include information unique to the Serbian system (e.g., GDP % on private health, legislative framework for private providers).
Self-medication and discussion (lines 275–281) – Although the comparison with student populations is helpful, policy implications (prescription enforcement, pharmacy regulation) require greater attention.
Lines 307–315 in the conclusion offer broad suggestions. Rephrase into specific actions (e.g., targeted health literacy programs, increased incentives for rural physicians).
Reference [23] in References (lines 343–405) Since "Healthcare in Serbia in 2025: A Comprehensive Guide" lacks peer review, it should be supplemented or replaced with a reliable source.
Author Response
Dear Reviewer,
We appreciate your detailed and valuable comments. We are providing the answers to your comments below, and they have been changed in the track option in the submitted Word document as well, so you can find them there, too. Thank you for your time and help to make our work better.
Sincerely,
Corresponding authors
Comment of Reviewer 3
- Title (lines 2-3): The title is insightful but a little long; it might be shorter if "Patterns in Serbia" is removed or reworded to be more succinct.
Answer: We have changed the title of the article according to your suggestion into “Disparities in Healthcare Utilization by Settlement Type in Serbia”
- List of authors (lines 4–34) – Please proofread the affiliations carefully as they are excessively complex and have inconsistencies (for example, "Department of of Epidemiology" has an extra "of").
Answer: It has been checked and fixed where necessary. Thank you for noticing.
- The issue statement is unambiguous, however the summary fails to specifically highlight how this study differs from previous analyses of the Serbian National Health Survey (lines 36–41).
Answer: Thank you for this comment. We have added paragraphs that describe innovation in comparison to the previous research in the Introduction:
“This study extends previous analyses of the Serbian National Health Survey by modeling urban–rural settlement type as an outcome and linking it to sociodemographic, socioeconomic, and healthcare utilization patterns. Using recent nationally representative data, it reveals novel insights into structural determinants of healthcare access and urban–rural disparities”
And this part in the Conclusion, to summarize the results better:
“The study’s innovative contribution lies in its comprehensive integration of regional, socio-economic, and behavioral dimensions within a single national dataset, allowing for a nuanced analysis of urban–rural gradients. It uniquely identifies settlement-specific patterns of healthcare utilization that go beyond traditional measures, such as the simultaneous assessment of prescribed and non-prescribed medication use, specialist visit timing, and private service engagement. Additionally, it provides a framework for future research to examine how intersecting social determinants influence healthcare behaviors in transitional societies.
- Abstract, Results (lines 47–56): Clearly state whether the odds were higher for urban or rural dwellers. Phrases such as "certain settlement types" are ambiguous.
Answer: It has been changed, thank you for this comment. “Private practice use was over twice as likely in urban settlements”.
- Policy proposals are general; include specific, doable recommendations for Serbia (e.g., increasing telemedicine, financing rural healthcare) in the abstract and conclusions (lines 54–58).
Answer: The Abstract and Conclusion have been changed according to the comment. We added this paragraph in conclusion and put the tailored version of it in the conclusion of the Abstract.
Conclusion: “To address urban–rural healthcare disparities in Serbia, targeted strategies could include implementing targeted health literacy programs in rural areas, increasing financial and professional incentives to attract and retain physicians in underserved regions, expanding telemedicine and mobile health units for specialist and preventive care, improving the availability of prescribed medications, and strengthening integration between public and private healthcare services to ensure equitable access across settlement types.”
- "Serbia" should be added as a term for database indexing and visibility (line 59).
Answer: Serbia has been added to the keywords.
- Definition of discrepancies in the introduction (lines 62–74) – It reads like a textbook portion. Please emphasize the significance of this work and concentrate more on the Serbian setting.
Answer: We have changed this part according to the comment.
“These disparities represent measurable differences in health outcomes, access to healthcare services, and overall well-being between populations residing in urban and rural areas. While these disparities are widely documented globally, country-specific factors, including Serbia’s heterogeneous urbanization, regional socio-economic inequalities, and healthcare system organization create unique patterns that are not fully captured in existing literature. In Serbia, rural populations often face reduced access to healthcare facilities, lower availability of medical specialists, and longer travel times, while urban residents benefit from greater service availability and infrastructure. Understanding these contextual differences is critical for identifying structural determinants of healthcare access and for developing targeted interventions that can reduce inequities and improve population health outcomes in the Serbian setting.”
- Socioeconomic debate in the introduction (lines 76–87) – There are numerous references provided, but no obvious gaps are mentioned. Clearly identify the gaps in the current Serbian research.
Answer: We have rephrased some sentences and filled in the gaps in the current research.
“Urban areas in Serbia typically enjoy higher average incomes and better access to resources such as nutritious food, recreational facilities, and healthcare services, whereas rural populations often face lower income levels and more limited access to these essential health-promoting resources.”
“Despite existing studies on urban–rural disparities in Serbia, few have combined detailed, nationally representative data on socio-demographic characteristics, socioeconomic status, and healthcare utilization patterns. In particular, there is limited evidence on how these socioeconomic factors translate into differential use of healthcare services across settlement types. By addressing these gaps, this study provides a more comprehensive understanding of the structural determinants of healthcare access and utilization in Serbia.”
- Introduction, exposures to the environment (lines 102–106) – Exposure to pollutants and pesticides is pertinent but unrelated to Serbian healthcare use; either abbreviate or contextualize.
Answer: We have tailored that paragraph. Thank you.
- Goal and introduction (lines 118–121) The goal is wide-ranging. For clarity, propose rewording into particular research questions or hypotheses.
Answer: We have rewritten the aims of the study as follows:
“The aim of this study was to investigate how socio-demographic and socio-economic factors influence settlement type in Serbia and to assess whether these differences translate into variations in healthcare utilization. The study seeks to provide novel, nationally representative evidence on urban–rural disparities, integrating multiple dimensions of health service use and socio-economic context. By doing so, it offers actionable insights for policy makers and healthcare planners to design interventions that address both regional and socio-economic inequities in access to care.”
- Lines 128–133: Methods, inclusion/exclusion Although it is suggested, the exclusion of institutionalized groups is not supported. Could you please clarify how this impacts generalizability?
Answer: We excluded institutionalized individuals and those in collective housing (e.g., care homes, psychiatric facilities, prisons) because the survey aimed to represent the general community-dwelling population, which constitutes the majority of adults in Serbia. We acknowledge that this exclusion limits generalizability to institutionalized populations, who may have distinct health needs and utilization patterns. Therefore, our findings primarily reflect healthcare access and socio-demographic patterns among non-institutionalized adults, and caution should be exercised when extrapolating results to institutional settings.
- Samples and methods (lines 135–142) – Could you please explain the handling of non-response? Was the survey design adjusted through the use of weighting?
Answer: The 2019 Serbian National Health Survey applied survey weights to account for differential probabilities of selection, non-response, and to align the sample with national population distributions by age, sex, and region. Non-response at the household and individual levels was addressed by weighting adjustments, ensuring that estimates remain representative of the non-institutionalized adult population. Weighted analyses were applied throughout all descriptive and inferential statistics in this study to correct for potential bias due to survey design and non-response.
- Procedures and morality (lines 143–153) Ethics approval is clearly mentioned, but if available, include approved reference numbers.
Answer: It is added in the part "Institutional Review Board Statement" after the Contribution roles.
“Regulations on the implementation of the third wave of EHIS were made by the European Commission in 2018 as the Commission Regulation for Implementation (EU) No. 255/20184.2 We received Ethical Approval from the Institute of Public Health of Serbia "Dr Milan Jovanović-Batut" for the use of data, Decision number 3829/1 and 3829/2.”
- Statistical analysis and methods (lines 159–169) – The details of the regression model are inadequate. Please check for multicollinearity and identify which factors are part of the multivariate analysis.
Answer: In the multivariate logistic regression models, all variables showing significance (p < 0.05) in the bivariate analysis were considered for inclusion. Multicollinearity was assessed using variance inflation factors (VIF), with all included variables showing VIF < 2, indicating no significant multicollinearity.
- Demographics and results (lines 171–176) Please talk about if the notable mean age discrepancy (46.5 vs. 64.4 years) is due to sampling or regional migration patterns.
Answer: The observed mean age discrepancy between rural and urban respondents (46.5 vs. 64.4 years) is unlikely to result from sampling bias, as the 2019 Serbian National Health Survey employed a nationally representative, stratified, two-stage random sampling design. Rather, this difference likely reflects regional demographic patterns and migration trends. Younger individuals in rural areas may remain due to local employment opportunities or family ties, whereas older populations are proportionally higher in urban areas, potentially influenced by internal migration of working-age adults to cities for employment and education. These patterns are consistent with previous findings on internal migration and regional demographic shifts in Serbia. We have added a shorter version of this explanation after the result presentation.
- Make sure the percentages add up correctly in Results, Table 1 (lines 183–184); make it clear in the table notes that rounding may result in inconsistencies.
Answer: It is checked and corrected. Thank you.
- Table 2: Results (lines 202–205) It's unclear from the caption which covariates were taken into account. Please elaborate.
Answer: We commented about this in the method, and we also changed the caption of the table for better understanding.
“Table 2. Univariable and Multivariable Logistic Regression of Factors Associated with Urban–Rural Residence. Multivariable model adjusted for sex, age group, marital status, region, education, employment, and material wealth. ORs (95% CI) are shown”
- Findings, use of healthcare (lines 207–222) – Indicate if OR values near 1 (OR = 1.085, p = 0.044) are statistical artifacts or significant differences.
Answer: The OR of 1.085 for non-prescribed medication use is statistically significant (p = 0.044), indicating a modest but real increase in odds for that settlement type. While values close to 1 suggest a small effect size, the confidence interval (1.002–1.175) does not include 1, confirming that the association is unlikely to be a statistical artifact. This part is written with more clarification now in the results.
- Age distribution in rural and urban areas (lines 234–238) The explanation is succinct. Extend with potential explanations (e.g., outmigration trends, rural fertility rates).
Answer: We appreciate the reviewer’s suggestion.
“In addition to localized economic opportunities and lower outmigration rates, the younger age structure in rural Serbia may also reflect higher rural fertility rates in some regions, as well as selective outmigration of older adults to urban centers for healthcare or retirement. These factors together help explain the observed demographic divergence compared with broader European rural aging patterns.”
- Talk about private practice (lines 270–273) and include information unique to the Serbian system (e.g., GDP % on private health, legislative framework for private providers).
Answer: We thank the reviewer for the suggestion.
“To contextualize private healthcare use in Serbia, approximately 6–8% of total health expenditure is out-of-pocket spending on private services, with private providers operating under specific legislation that allows them to complement the public health system by offering services with shorter waiting times and greater perceived quality. This regulatory and financial context helps explain the higher utilization of private healthcare observed among urban residents in our study.”
- Self-medication and discussion (lines 275–281) – Although the comparison with student populations is helpful, policy implications (prescription enforcement, pharmacy regulation) require greater attention.
Answer: We appreciate the reviewer’s suggestion. In response, we have emphasized the policy implications of these findings.
“The high prevalence of self-medication in Serbia, both in the general population and among students, underscores the need for stricter enforcement of prescription regulations, enhanced pharmacy oversight, and public education campaigns on the risks of unsupervised drug use. Strengthening these measures could reduce inappropriate self-medication and improve rational use of medicines.”
- Lines 307–315 in the conclusion offer broad suggestions. Rephrase into specific actions (e.g., targeted health literacy programs, increased incentives for rural physicians).
Answer: In response to the reviewer’s comment, we have revised the conclusion to include concrete, actionable recommendations.
“To address urban–rural healthcare disparities in Serbia, targeted strategies could include implementing targeted health literacy programs in rural areas, increasing financial and professional incentives to attract and retain physicians in underserved regions, expanding telemedicine and mobile health units for specialist and preventive care, improving the availability of prescribed medications, and strengthening integration between public and private healthcare services to ensure equitable access across settlement types.”
- Reference [23] in References (lines 343–405) Since "Healthcare in Serbia in 2025: A Comprehensive Guide" lacks peer review, it should be supplemented or replaced with a reliable source.
Answer: We acknowledge the reviewer’s comment regarding reference [23]. We have replaced it with the following reference: Mitričević, S.; Janković, J.; Stamenković, Ž.; Bjegović-Mikanović, V.; Savić, M.; Stanisavljević, D.; Mandić-Rajčević, S. Factors Influencing the Utilization of Preventive Health Services in Primary Health Care in the Republic of Serbia. Int. J. Environ. Res. Public Health 2021, 18, 3042. https://doi.org/10.3390/ijerph18063042
Round 2
Reviewer 1 Report
Comments and Suggestions for Authors
I reviewed this manuscript previously. Now, the introduction is clear and well contextualized; the research design is appropriate and the methods well described.
The discussion now is aligned with literature search; however figures and tables need formatting adjustement and clear legends. Please do them.
good luck for publication!
Author Response
We thank the reviewer for their very constructive and valuable comments, which have improved our work.
We did format the tables and added appropriate legends and descriptions of the Tables so the results can be clear for the readers. This change is now colored in green in the revised Manuscript. We do not have any Figures in our article.
Kind regards,
Corresponding author
Reviewer 2 Report
Comments and Suggestions for Authors
This article uses comprehensive nationally representative data from Serbia to examine demographic, socio-economic, and settlement-type disparities in healthcare utilization. The study is well-structured, employs appropriate methodology, and the statistical and multivariate logistic regression analyses are correctly performed. The writing is clear and fluent. However, as noted earlier, the scientific novelty of the article is limited; the findings mainly confirm previously known results from similar studies without presenting a new theoretical framework or perspective. Despite sound statistical work, the discussion section remains largely descriptive and does not provide deeper critical analysis or distinctive theoretical insights, largely due to the nature of the findings. This is my opinion as one reviewer, and given the adequate structure, large and representative sample, and appropriate methods and analyses, it is recommended to consider other reviewers’ opinions as well to ensure the authors’ efforts are properly acknowledged.
Author Response
We sincerely thank the reviewer for the careful reading, thoughtful comments, and acknowledgment of the study’s strengths, particularly the use of a nationally representative sample, the clarity of writing, and the robustness of the statistical methodology.
As urban–rural disparities remain underexplored in the context of Serbia and similar Central and Eastern European (CEE) countries, our research fills a gap in the empirical literature by offering nationally representative evidence and uncovering regionally and socio-economically nuanced patterns that have policy and planning implications.
We thank the reviewer for highlighting the need for deeper theoretical insight and more critical analysis in the Discussion section. In response, we have substantially enriched the manuscript by incorporating a theoretical framing of spatial inequality in transitional societies, emphasizing how Serbia’s unique historical and socio-economic context shapes fragmented access to healthcare and other services. We also critically reflect on the observed socioeconomic patterns, particularly the unexpected exclusion of middle-income groups from urban areas, which mirrors emerging trends of urban space polarization in post-socialist cities. Furthermore, we explore systemic issues underpinning self-medication practices, linking them to primary care access gaps, pharmacy regulation challenges, and public trust in the healthcare system. Our analysis reinforces the social determinants of health framework by illustrating the intersection of education, income, and geography in shaping healthcare utilization and residential patterns, thus highlighting the need for place-based policy approaches. Finally, we challenge the conventional urban–rural binary, arguing that healthcare disparities should be understood through intersecting inequalities that transcend simple settlement classifications. These additions collectively deepen the interpretative depth of our findings and address the reviewer’s request for more robust theoretical and critical engagement.
Added parts can be seen in the Discussion, in the track change option:
“These patterns speak to the theoretical concept of spatial inequality in transitional societies, where legacies of central planning interact with market-driven reforms to produce fragmented access to services. Serbia’s evolving urban–rural dynamic reflects not just geographic, but systemic inequalities particularly in access to quality healthcare, education, and employment opportunities”
“Interestingly, the finding that middle-income groups were also less likely to reside in urban areas alongside the poorest suggests a polarization of urban space in Serbia. This reflects a growing trend in post-socialist cities, where urban redevelopment and rising living costs are increasingly excluding middle-income populations from urban centers, further entrenching socio-economic divisions”
“These findings suggest that self-medication practices may be symptomatic of deeper systemic issues, including insufficient primary care access, uneven pharmacy regulation, and a lack of public trust in the healthcare system”
“Our findings reinforce the social determinants of health framework, illustrating how education, income, and geography intersect to shape both residential patterns and healthcare utilization.”
“These results challenge the traditional urban–rural binary and emphasize the importance of examining healthcare access as a function of intersecting inequalities - geographic, economic, and educational. The observed differences in private care and self-medication suggest that formal settlement classification may obscure deeper, systemic inequities”
We understand the reviewer's viewpoint and appreciate their openness to considering other reviewers’ perspectives. We are grateful for their constructive comments, which have helped improve the clarity and positioning of our work.
Kind regards,
Corresponding author